# Comparative Study of CDST & Multiplex PCR to Detect MBL Producing Gram-Negative Bacilli among VAP Patients Admitted in a Public Medical College Hospital of Bangladesh

**DOI:** 10.3390/pathogens8030151

**Published:** 2019-09-12

**Authors:** Tanzina Nusrat, Nasima Akter, Mainul Haque, Nor Azlina A. Rahman, Arup Kanti Dewanjee, Shakeel Ahmed, Diana Thecla D. Rozario

**Affiliations:** 1Department of Microbiology, Chattogram Medical College, Chattogram 4217, Bangladesh; tanzina.nusrat@gmail.com (T.N.); nasima196177@yahoo.com (N.A.); 2Unit of Pharmacology, Faculty of Medicine and Defence Health, Universiti Pertahanan Nasional Malaysia, (National Defence University of Malaysia), Kem Sungai Besi, Kuala Lumpur 57000, Malaysia; 3Department of Basic Health, Kulliyyah of Allied Health Sciences, International Islamic University Malaysia, Jalan Sultan Ahmad Shah, Bandar Indera Mahkota, Kuantan 25200, Malaysia; nazara@iium.edu.my; 4Department of Microbiology, Marine City Medical College, Chattogram 4217, Bangladesh; arupdewanjee@yahoo.com; 5Department of Microbiology, Bangladesh Institute of Tropical & Infectious Diseases, Chattogram 4217, Bangladesh; shakeelcmc@gmail.com; 6Department of Microbiology, Colonel Malek Medical College, Manikgonj 6700, Bangladesh; dianadrozario17@yahoo.com

**Keywords:** VAP, MBL, multiplex PCR, metallo-β-lactamases, CDST test, bronchoalveolar lavage, BAL

## Abstract

Background: Ventilator-associated pneumonia (VAP) is the most common nosocomial infection in intensive care units (ICU), which accounts for 25% of all ICU infection. Documenting carbapenem-resistant gram-negative bacilli is very important as these strains may often cause outbreaks in the ICU setting and are responsible for the increased mortality and morbidity or limiting therapeutic options. The classical phenotypic method cannot provide an efficient means of diagnosis of the metallo-β-lactamases (MBLs) producer. Polymerase chain reaction (PCR) assays have lessened the importance of the phenotypic approach by detecting metallo-β-lactamase resistance genes such as New Delhi metallo-β-lactamase (NDM), Imipenemase (IMP), Verona integron-encoded metallo-β-lactamase (VIM), Sao Paulo metallo-β-lactamase (SPM), Germany Imipenemase (GIM). Objective: To compare the results of the Combined Disc Synergy Test (CDST) with that of the multiplex PCR to detect MBL-producing gram-negative bacilli. Materials and Method: A total of 105 endotracheal aspirates (ETA) samples were collected from the ICU of a public school in Bangladesh. This cross-sectional study was carried out in the Department of Microbiology, Chittagong for quantitative culture, CDST test, and multiplex PCR for bla_IMP_, bla_VIM_, bla_NDM_ genes of MBL producers. Results: Among the 105 clinically suspected VAP cases, the quantitative culture was positive in 95 (90%) and among 95 g-negative bacilli isolated from VAP patients, 46 (48.42%) were imipenem resistant, 30 (65.22%) were MBL producers by CDST, 21 (45.65%) were identified as MBL producers by multiplex PCR. Conclusion: PCR was highly sensitive and specific for the detection of MBL producers.

## 1. Introduction

VAP is defined as pneumonia occurring after the first 48 h of starting mechanical ventilation [1,2]. VAP is often correlated with an extension of the duration of hospital stay that necessitates Intensive Care Unit (ICU), mechanical ventilation (MV), subsequently, increasing healthcare overheads and mortality [2,3]. The VAP related mortality rate varies from 20–75% [4]. The documentation of the causative microorganism of VAP and their antimicrobial sensitivity is the fundamental issue for the overall treatment planning and to ensure better treatment outcome [5,6]. Protected specimen brush (PSB) or bronchoalveolar lavage (BAL), or open-lung biopsy, have been established in detecting VAP contributory microorganisms and commonly measure higher values than endotracheal aspirates (ETA) [7,8]. The ETA gram-staining, nonquantitative, and semiquantitative culture method is dependable and reliable requiring small procedural proficiency but no specific apparatus [8,9]. Quantitative ETA culture (QETAC) is a non-invasive method that is cheaper than quantitative BAL fluid culture, and QETA has the potential to substitute for BAL in quantitative cultures [10]. The optimum level frequently embraced for detecting and establishing pneumonia by quantitative cultures are ≥10^5^ to 10^6^, 10^4^, and 10^3^ CFU/mL for QETAC, bronchoscopic BAL, and PSB, correspondingly, with 10^5^ CFU/mL being the most extensively recognized value [7,9,11,12]. The emergence of antimicrobial resistance infection is becoming a significant health problem worldwide, especially in healthcare-associated infections (HCAIs) [2,13]. Carbapenem-resistant gram-negative bacilli are very important as these strains frequently cause outbreaks in the ICU setting and are responsible for the increased mortality and morbidity and limit therapeutic options [14,15]. Carbapenemase, namely the New Delhi MBLs, are producing microorganisms emerging rapidly throughout the globe and leading to enormous public health threat [16,17]. Rapid detection of these MBLs is necessary to institute appropriate treatment and effective infection control measures [17,18]. The classical phenotypic method cannot provide an efficient means of diagnosis of MBL producers because most of the atypical bacteria grow either slowly or not in culture, leading to delays in the detection and diagnosis [19,20]. Multiplex PCR assays able to detect MBL resistance genes, such as the NDM, IMP, VIM, SPM, GIM, consequently, the importance of the old-age classical phenotypic method was reduced in medical science [21,22]. MBLs are rapidly emerging resistance elements among *Pseudomonas aeruginosa* and other gram-negative pathogens [23]. Individually, the MBL gene is situated on precise genetic elements, including integrons, transposons, plasmids, or on the chromosome, in which they transmit genes encoding factors of resistance to carbapenems and other antibiotics, pondering multidrug resistance [24]. After that, the proper diagnosis of causative microorganisms is essential.

This study was designed to detect causative microorganism VAP. Initially, isolation and identification of gram-negative bacteria by quantitative culture methods was performed and the determination of MBL-producing bacteria by phenotypic methods (CDST) and the confirmation of MBL genes by multiplex PCR.

## 2. Objectives of the Study

To compare the results of CDST with that of the multiplex PCR to detect MBL-producing gram-negative bacilli.

## 3. Materials and Methods

Study design, place period, and population: This cross-sectional study was conducted in the department of Microbiology of Chattagram Medical College from July 2017 to June 2018. A total of 105 suspected VAP patients admitted to the ICU of Chattagram Medical College Hospital (CMCH) of age as ranging from 5 to 75 years were included in the study. Sample selection: The patients who had mechanical ventilation by endotracheal tube for more than 48 h along with two or three of the following criteria suggesting VAP: (i) Fever/hypothermia or leukocytosis/leucopenia; (ii) Purulent tracheal discharge; (iii) Positive chest X-ray (chest X-ray shows consolidation or infiltration or pleural effusion) were included in the study. Patients who had severe hypoxemia (PaO_2_/FiO_2_ < 100), immunocompromised, or neutropenic symptoms were excluded from the study. The selection criteria were based on the earlier published study [25]. Data collection: Data was collected, recorded, edited, and analyzed in a predesigned datasheet. Statistical analysis: The results of the experiments were recorded and analyzed systematically using descriptive analysis. The diagnostic values of CDST against PCR as the gold standard (sensitivity, specificity, predictive values, accuracy, and their 95% confidence interval (CI)) were analyzed using MEDCALC^®^, an online software [26]. SPSS Software (IBM Corp. Released 2016. IBM SPSS Statistics for Windows, Version 24.0. IBM Corp.: Armonk, New York, NY, USA) was used to analyze the comparability of the two tests using McNemar’s test for dependent variables. The significance value was set to 0.05. Ethical approval: This study obtained ethical approval from Chattagram Medical College, Chattagram, Bangladesh (Reference No.: CMC/PG/2017/322, Date: 04-05-2017).

### Laboratory Method

This study intended to identify bla_IMP_, bla_NDM_, and the bla_VIM_ genes of β-lactamase producing gram-negative bacteria. A total of 105 ETA samples were collected by mucus extractors in a sterile capped container and immediately brought to the laboratory. Standard methods were used for the analysis and culture of ETA specimens collected from all suspected patients. Immediately after receipt, one mL of ETA was diluted to a final concentration of 1:100 in sterile phosphate buffer solution. After being manually stirred, a sample was taken with the help of 0.01 mL calibrated loop and was cultured on 5% sheep blood agar and chocolate agar, and MacConkey’s agar plates then incubated in 5% CO_2_ at 35 °C for 48–72 h. The cut-off point for the ETA was 10^5^–10^6^ cfu/mL for the quantitative culture. All isolates were identified based on their colony morphology, culture characteristics, and biochemical reactions according to the standard microbiological procedures. The imipenem resistant screening test was performed by the Kirby–Bauer disc diffusion test. The media used was Mueller–Hinton Agar Media, the potency of the imipenem disc was 10 µg. The standard values determined by the National Committee for Clinical Laboratory Standards (NCCLS) pathogens were classified as sensitive (21 mm), intermediate resistant (16–20 mm), and resistant (15 mm) [27,28] and a zone of inhibition of ≥20 mm was considered sensitive, a zone of inhibition less than 20 mm was considered resistant to Imipenem [29]. All the imipenem-resistant *Pseudomonas, E. coli, Klebsiella*, and *Acinetobacter*, were tested for detection of MBLs by Imipenem-EDTA CDST. ETA was preserved at −70 °C for DNA extraction. Test organisms were inoculated onto plates with Muller–Hinton agar as recommended by CLSI, 2019. Two 10 μg imipenem discs (Becton Dickinson, Franklin Lakes, NJ, U.S.) were placed on the plate, and appropriate amounts of 10 μl of Ethylenediaminetetraacetic acid (EDTA) solution was added to one of them to obtain the desired concentration (750 µg). The inhibition zones of the imipenem and imipenem-EDTA discs were compared after 16 to 18 h of incubation at 37 °C. In the Combined Disc Test, the increase in the inhibition zone with the imipenem and EDTA disc was >7 mm than the imipenem disc alone, it was considered as MBLs positive [30,31,32,33,34]. The PCR assay was performed on culture isolates using the specific primers to detect the blaIMP, blaIMP, and blaNDM genes [35,36]. DNA from imipenem-resistant culture isolates was extracted according to the GeneJet DNA extraction kit (ThermoFisher Scientific, Waltham, MA, USA). Primer sequence used for amplification was blaIMP (F: GGAATAGAGTGGCTTAAYTCTC, R: GGTTTAAYAAAACAACCACC), blaVIM (F: GATGGTGTTTGGTCGCATA, R: CGAATGCGCAGCACCAG), blaNDM (F: GGTTTGGCGATCTGGTTTC, blaNDM-R: CGGAATGGCTCATCACGATC) [37]. The composition of the PCR Master Mix was 0.05 U/ µL TaqDNA polymerase; reaction buffer was 4 mM MgCl2, 0.4 mM of each dNTP (dATP, dCTP, dGTP, dTTP) (ThermoFisher Scientific, Waltham, MA, USA). The final reaction volume was 50 µL. The amplification (PCR) was performed in a thermal cycler (Eppendorf AG, Hamburg, Germany). The protocol of thermal cycles of PCR for the detection of the genes included the initial denaturation at 95 °C for 3 min-1 cycle, denaturation at 95 °C for 30 s, primer annealing at 55 °C for 30 s, extension at 72 °C for 20 s-35 cycles, final extension at 72 °C for 10 min-1 cycle (ThermoFisher Scientific, Waltham, MA, USA). The final amplified products were subjected to electrophoresis on 1.5% agarose gel. The presence of the bla_IMP_-*232*, bla_VIM_-390, and bla_NDM_-621 band under UV transilluminator showed positive results.

## 4. Results

A total of 105 clinically suspected VAP cases were enrolled in this study from CMCH. The mean age of the patients was 51.53 ± 18.81 years. Ninety-five (90%) clinically suspected VAP cases were positive for the quantitative culture method (Figure 1). The highest number (30, 31.58%) of culture-positive cases were in the age group of 66 to 75 years (Table 1). Among the 95 culture-positive cases, 69 (72.63%) were male, and 26 (27.37%) were female where the male-female ratio was 2.65:1. MBL- producing gram-negative bacteria were identified by the imipenem-resistant screening test (Table 2). Among 95 gram-negative bacilli isolated from VAP patients, 46 (48.42%) were imipenem resistant. Out of the 46 IMP-screened positive cases, 23 (56.1%) were *Acinetobacter* spp. followed by 13 (54.17%) *Klebsiella* spp., 6 (33.33%) *Pseudomonas* spp., 3 (33.33%) *E. coli*, and one (25%) *Proteus* spp. Out of the 46 imipenem-resistant positive gram-negative bacteria, 30 (65.22%) were MBL-producers by CDST (Figure 2). Twenty-one (45.65%) were identified as MBL-producers by multiplex PCR (Figure 3 and Figure 4). Snapshots showing the multiplex PCR show gel electrophoresis of amplified DNA of bla_IMP_ and bla_NDM_ (Figure 4) but fail to identify the bla_VIM_ gene. The results of the CDST test were compared with PCR as the gold standard by Chi-square tests (Table 3). The difference between the CDST and PCR to detect MBL-producers was statistically significant (*p* < 0.05).

Table 4 shows the diagnostic values of CDST against PCR as the gold standard. The sensitivity (true positive rate) was quite high (85.7%) due to high positive rates of CDST, but unfortunately, this leads to high false-positives leading to lower specificity (true negative rate, 52.0%). With the accuracy of 67.4%, the McNemar’s test performed showed a significant difference between the results of CDST and PCR with the *p* value of 0.035.

## 5. Discussion

Lower respiratory tract infection (LRTI) is a well-recognized complication of artificial ventilation and is a common cause of morbidity and mortality in the ICU [2,38]. The snowballing frequency of the multidrug-resistant microorganisms’ in ICU calls for quick and precise diagnosis of VAP, which is essential for the ideal antimicrobial intervention [5]. In the present study, quantitative culture was utilized, and it was 90% positive. The CDC (Center for Disease Control) also recommends quantitative culture methods for respiratory and VAP specimens [39]. Although another study reported that in diagnosing VAP, quantitative cultures were of limited value when utilizing endotracheal aspirates [40]. The quantitative culture of lower respiratory tract specimens obtained by non-bronchoscopic procedures is hugely beneficial in the quick diagnosis and proper therapeutic intervention of VAP [41]. There were multiple studies which showed that the high percentage of positive quantitative cultures regarding VAP patient’s management [42,43].

The gender variation of the study findings was similar to another published study [44], and multiple studies reported that males were predominant suffers from VAP (46,47) that were in the same line of the current study [44,45,46]. Male predominance could be explained by the fact that Bangladesh has a large working community composed mainly of males. Multiple studies reported that among several factors, some are not changeable. These include preexisting pulmonary disease, AIDS (acquired immune deficiency syndrome), coma, head trauma, and multiple organ system failures, age of over 60 years, acute respiratory distress syndrome, chronic obstructive pulmonary disease, and male sex [47,48,49].

MBLs producing gram-negative bacteria were identified by the imipenem-resistant screening test in the present study, and 48.42% were imipenem-resistant. One recent Indian study reported that 28.04% were resistant to imipenem [50]. After that, the current study was higher than the study conducted in India [50]. Again, 65.22% and 45.65% imipenem-resistant MBLs were identified among the positive screening of gram-negative bacteria CDST and multiplex PCR, respectively. The earlier-mentioned Indian study reported that 70% of the imipenem-resistant gram-negative bacteria were CDST positive. Additionally, another study revealed that 65% were MBL-producers by the multiplex PCR method, which was higher than the present study [51].

The difference between CDST and PCR to detect MBL-producers was statistically significant (*p* < 0.05). CDST identified more cases of negative findings of positive imipenem than the multiplex PCR in the current study. Although CDST is a phenotypic test to detect MBL-generating organisms with the utmost sensitivity and specificity; nevertheless, multiple studies reported high false-positive results, not only for CDST but all phenotypic analysis [31,33,52,53]. Primary resistance mechanisms responsible for resistance to carbapenems is because of the production of different types of carbapenemase enzymes, namely MBL, KPC, or Oxacillinases (OXA) enzyme [17,54]. The ethylenediaminetetraacetic acid (EDTA) or MPA is utilized in CDST to possibly chelate and inhibit OXA enzymes, after that, producing a result that could be interpreted as a false-positive for the presence of MBL [55]. One Bangladeshi study reported that carbapenemase genes were spotted in 11.6% among gram-negative pathogens in urinary tract infections and entirely gram-negative microorganisms’ community were carbapenemase-producing and were resistant to colistin. This study revealed that a significant rate of urinary isolates was carbapenemase producers, including a high prevalence of blaNDM-1, in Bangladesh. One Bangladeshi study reported that the carbapenemase gene, blaNDM-1, was spotted in 11.6% of gram-negative pathogens in urinary tract infections and entirely gram-negative microorganisms’ community were carbapenemase-producing were and resistant to colistin [56].

One more study also reported that the presence of New Delhi metallo-ß-lactamase-1 (NDM-1) producing E. coli in Bangladesh [57]. Another study conducted in a tertiary diabetic hospital situated in the capital city of Bangladesh reported that pus, blood, urine, and tracheal aspirates derived Pseudomonas isolates where the majority (81.8%) were resistant to imipenem [58]. Clinical specimens, such as endotracheal, throat, tracheal aspirates, sputum, catheter tips, urine, pus, and wounds derived 3.5% of the gram-negative microorganisms were NDM-1 produced in Bangladesh [59]. It has been reported from the neighboring country, Burma owns three discrete *bla*_NDM_-fostering plasmids among carbapenemase-producing *Enterobacteriaceae* [60]. Another Indian study conducted in neonatal ICU (NICU) utilizing a PCR-based investigation identified pathogens containing NDM-1, NDM-4, NDM-5, NDM-7, along with OXA, CMY, and SHV variants on conjugative plasmids of IncFIA, IncFIC, IncF, IncK, IncFIB, IncB/O, IncHI1, IncP, IncY, IncFIIA, IncI1, and IncN types. Subsequently, carbapenem-resistant NDM-producing Enterobacteriaceae isolates were recovered from NICU [61]. Another study revealed that carbapenemase NDM-1-producing *Enterobacteriaceae* is a substantial burden of public health in the Indian subcontinent [62]. One more study examined endotracheal aspirates among patients of ICU reported that 96% of *Acinetobacter baumannii* were multidrug-resistant. The AMR (Antimicrobial resistance) genes such as *bla_ampC_* and carbapenemase genes (*bla_MBLs_* and *bla_KPC_*) were found in 95.83%, and 100% of *A. baumanni* were cephalosporin and imipenem resistant, respectively [63]. After that, the current study showed the MBL-producing gram-negative microorganism presence through multiplex PCR among VAP patients were most likely. There are several MBL-producing genes, but the current utilizes only three *bla*_NDM_, *bla_I_*_MP_, *bla*_VIM_ genes because of financial constraints.

## 6. Conclusions

The problem of phenotypic detection of MBL by CDST is that the high false-positive reporting rate exists in the current study. So multiplex PCR is recommended for optimal detection of MBL-producers, at least in the tertiary care hospitals of Bangladesh.

## 7. Recommendations

The prevention of antimicrobial resistance in ICU patients should focus on recognition via routine unit-based surveillance, improved compliance with handwashing and barrier precautions, and antibiotic-use policies tailored to individual units within hospitals. Carbapenems use should be restricted. More research has been recommended in ICUs in Bangladesh to prevent AMR and to minimize the resistant microbial infection.

## Figures and Tables

**Figure 1 pathogens-08-00151-f001:**
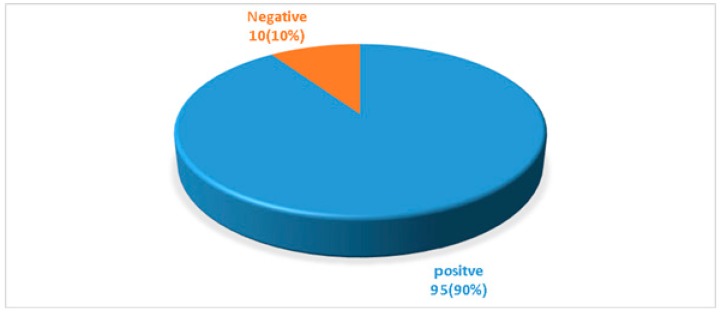
Pie Chart is showing the rate of isolation of bacteria from clinically suspected VAP cases by quantitative culture. Total cases = 105.

**Figure 2 pathogens-08-00151-f002:**
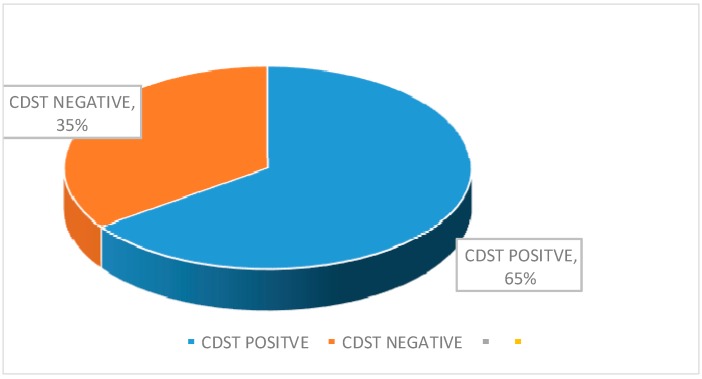
Pie chart showing the rate of detection of MBLs producing gram-negative bacilli by CDST.

**Figure 3 pathogens-08-00151-f003:**
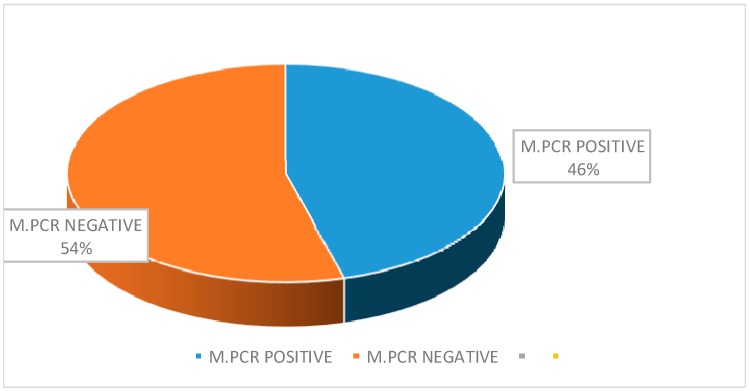
Pie chart showing the percentage of detection of MBLs producing gram-negative bacilli by multiplex PCR.

**Figure 4 pathogens-08-00151-f004:**
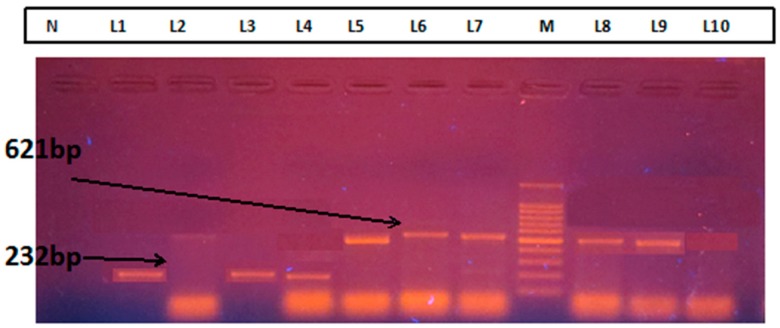
Snapshot is showing multiplex PCR gel electrophoresis of amplified DNA. **N**–Negative Control, **M**–DNA ladder, **L**1,3,4–showing a positive band of the bla_IMP_ gene, **L**5,6,7,8,9–indicating a definite band of the bla_NDM_ gene, **L**2–no gene detected, **L**10-is the positive control.

**Table 1 pathogens-08-00151-t001:** Age and Sex distribution of culture-positive VAP cases (*n* = 95).

Age Group (Years)	Sex (%)		Suspected VAP Cases
	Male	Female	
5–15	3	1	4
16–25	5	4	9
26–35	6	4	10
36–45	6	1	7
46–55	11	4	15
56–65	18	2	20
66–75	20	10	30
Total	69 (72.63%)	26 (27.37%)	95

Male: Female = 2.65:1. Mean ± SD = 51.53 ± 18.81.

**Table 2 pathogens-08-00151-t002:** The distribution of MBL-producing gram-negative bacilli by the imipenem-resistant screening test (*n* = 95).

Name of the Organism	Imipenem-Resistant Screening Test (%)	
	Positive	Negative
*Acinetobacter* spp. (41)	23 (56.1)	18
*Klebsiella* spp. (24)	13 (54.17)	11
*Pseudomonas* spp. (18)	6 (33.33)	12
*E. coli* (9)	3 (33.33)	6
*Proteus* spp. (4)	1 (25)	3
Total	46 (48.42)	49 (51.60)

**Table 3 pathogens-08-00151-t003:** Comparison of CDST and multiplex PCR considering multiplex PCR as the gold standard (n = 46).

Test		PCR		Total
		Positive	Negative	
**CDST**	**Positive**	18	12	30
	**Negative**	3	13	16
Total		21	25	46

Χ2 value = 7.1. *p* < 0.05.

**Table 4 pathogens-08-00151-t004:** Diagnostic values of CDST against PCR as the gold standard.

Statistic	Value	95% Confidence Interval
Sensitivity	85.7%	63.7–97.0%
Specificity	52.0%	31.3–72.2%
Positive predictive value	60.0%	49.0–70.0%
Negative predictive value	81.3%	58.7–93.0%
Accuracy	67.4%	52.0–80.5%

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
