# Peer review of "Comparative Study of CDST & Multiplex PCR to Detect MBL Producing Gram-Negative Bacilli among VAP Patients Admitted in a Public Medical College Hospital of Bangladesh"

_pathogens, 2019, doi:10.3390/pathogens8030151_

Round 1
Reviewer 1 Report
This is an interesting manuscript comparing results of CDST with multiplex PCR to detect MBL producing gram-negative bacilli. The authors reported that PCR was highly sensitive and specific for detection of MBL producers. While the paper overall is well-written, the results are not conclusive given the unadjusted statistical methods. I would suggest that the authors consult a statistician before resubmitting their manuscript. Some minor English language editing also is required. The limitations paragraph needs further development.
Author Response
Comments and Suggestions for Authors
This is an interesting manuscript comparing results of CDST with multiplex PCR to detect MBL producing gram-negative bacilli. The authors reported that PCR was highly sensitive and specific for detection of MBL producers. While the paper overall is well-written, the results are not conclusive given the unadjusted statistical methods. I would suggest that the authors consult a statistician before resubmitting their manuscript. Some minor English language editing also is required. The limitations paragraph needs further development.
Analysis done again with proper statistics expert and adjusted result incorporated with a new Table 4. English language has thoroughly edited by Dr MS Razzaque, Professor of Pathology, Lake Erie College of Osteopathic Medicine, 1858 West Grandview Boulevard Erie, PA 16509, USA.
Reviewer 2 Report
Dear authors of the "Comparative Study of CDST & Multiplex PCR to 2 Detect MBL Producing Gram-negative Bacilli among 3 VAP Patients in Chittagong Medical College Hospital" manuscript.
Your study is trying to address an important topic, and you can improve your paper by indicating the importance of your study in the field of health studies, using clear labeling and figure discription, improve data presentation.
Here are some comments to address:
line 45-46: re-write the sentence isn't clear.
Line 66: add more information about these resistance genes: NDM, IMP, VIM, SPM, GIM (such are what they stand for, in what bacteria strains are found? ..etc.)
lines 68 -73: need to state the aim of the study precisely and avoid redundancy.
line 77: correct "from 5- > 65 years" to "from 5 to 65 years"
line 82 -83: but why these criteria were selected in that study?
lines 83-85: what software/programme was used for data analysis?
line 132: correct "group of 66- < 75 years" to "between 66 to 75 year"
line 150: dose the chare shoes rate or percentage? read throughout the manuscript and make sure to use the correct statistical term.
Fig 1: use clear labeling, needs more information to indicate what 95 refers to and what 10 is?
Fig 2: need to improve the quality and the labeling.
Table 1 and 2: use the symbol "%" next to the percentage to make it easier for the reader to distinguish between numbers and percentages.
-The PCR data is not clear. Need to show PCR results for the specifically mentioned resistant genes.
- Also, need to add figures of the gels with the genes bands in the appendix. Such data must be available to show the accuracy of the work.
Thank you,
Author Response
Here are some comments to address:
line 45-46: re-write the sentence isn't clear.
Altered.
Line 66: add more information about these resistance genes: NDM, IMP, VIM, SPM, GIM (such are what they stand for, in what bacteria strains are found? ..etc.)
Additional information incorporated.
lines 68 -73: need to state the aim of the study precisely and avoid redundancy.
Altered
line 77: correct "from 5- > 65 years" to "from 5 to 65 years"
Corrected
line 82 -83: but why these criteria were selected in that study?
Criteria was adopted from earlier published study
lines 83-85: what software/programme was used for data analysis?
Software details mentioned
line 132: correct "group of 66- < 75 years" to "between 66 to 75 year"
Corrected
line 150: dose the chare shoes rate or percentage? read throughout the manuscript and make sure to use the correct statistical term.
Fig 1: use clear labeling, needs more information to indicate what 95 refers to and what 10 is?
Correction done
Fig 2: need to improve the quality and the labeling.
Corrected
Table 1 and 2: use the symbol "%" next to the percentage to make it easier for the reader to distinguish between numbers and percentages.
Both corrected accordingly
-The PCR data is not clear. Need to show PCR results for the specifically mentioned resistant genes.
Table 4 to clarify
- Also, need to add figures of the gels with the genes bands in the appendix. Such data must be available to show the accuracy of the work.
Figures Added.
Reviewer 3 Report
Review for Manuscript pathogens-578663-peer-review-v1
General Comments: Initially, while the authors are addressing an important clinical question – comparison of CDST to PCR to detect MBL producing bacteria, significant changes must be made to the manuscript to help the reader and reviewer.
The title would be better if not referencing the institution. In the abstract, indicating that it was in the Department of Pathology, Chittagong is not necessary here. In the materials and methods, were the patients 5-65 years of age? It appears to be “5 to greater than 65 years of age”. Perhaps write out 5 to 65 so symbols do not become a problem. Statistical significance level needs to be reported for the statistical analysis section (P < 0.05). The materials and methods section need defined subheadings, for example “Patient Inclusion” “Microbiology” “PCR” “Statistical Analysis”, etc. This will help the reader and reviewer. Ethical approval should be under Patient Inclusion in the materials and methods. Figure 1 – Brackets are empty and what do they mean? Table 2 – Should parenthesis be around the numbers by genus spp like with Acinetobacter? The discussion section needs to be divided into paragraphs. The limitations section needs to be more than a sentence. The limitations and recommendations section should be made as part of the discussion. The manuscript needs to be corrected by an English editor or proofreading service.Author Response
Here are some comments to address:
line 45-46: re-write the sentence isn't clear.
Altered.
Line 66: add more information about these resistance genes: NDM, IMP, VIM, SPM, GIM (such are what they stand for, in what bacteria strains are found? ..etc.)
Additional information incorporated.
lines 68 -73: need to state the aim of the study precisely and avoid redundancy.
Altered
line 77: correct "from 5- > 65 years" to "from 5 to 65 years"
Corrected
line 82 -83: but why these criteria were selected in that study?
Criteria was adopted from earlier published study
lines 83-85: what software/programme was used for data analysis?
Software details mentioned
line 132: correct "group of 66- < 75 years" to "between 66 to 75 year"
Corrected
line 150: dose the chare shoes rate or percentage? read throughout the manuscript and make sure to use the correct statistical term.
Fig 1: use clear labeling, needs more information to indicate what 95 refers to and what 10 is?
Correction done
Fig 2: need to improve the quality and the labeling.
Corrected
Table 1 and 2: use the symbol "%" next to the percentage to make it easier for the reader to distinguish between numbers and percentages.
Both corrected accordingly
-The PCR data is not clear. Need to show PCR results for the specifically mentioned resistant genes.
Table 4 to clarify
- Also, need to add figures of the gels with the genes bands in the appendix. Such data must be available to show the accuracy of the work.
Figures Added.
Round 2
Reviewer 2 Report
Dear authors,
Thank you for taking the time to edit the manuscript and do the requested changes.
Here are some notes:
Line 162: incorrect use of capitalization, and please keep it consistent all figure's descriptions. Figure 2 and 3: remove the additional symbols in the legends. Figure 4: labeling must be improved. Fig 4: what are lane 2 and 10? Must be mentioned in the description. Fig 4: shows bands just of 2 of the examined genes, is it possible to show the other genes as well? Need to mention figure 4 in the results section. Additional English editing is recommended. Figures and tables need to be improved in terms of labeling and description.Author Response
Our Response
Many Thanks.
Line 162: incorrect use of capitalization, and please keep it consistent all figure's descriptions.
Corrected
Figure 2 and 3: remove the additional symbols in the legends.
Removed.
Figure 4: labeling must be improved. Fig 4: what are lane 2 and 10? Must be mentioned in the description. Fig 4: shows bands just of 2 of the examined genes, is it possible to show the other genes as well? Need to mention figure 4 in the results section.
Efforts made to alter.
This study was out of pocket research. There was obviously financial constraint. Thereafter, intended to identify three genes. blaVIM was not identified. Therefore, unable to add other figures.
Additional English editing is recommended. Figures and tables need to be improved in terms of labeling and description.
Efforts to make to improve.
Reviewer 3 Report
The authors have nicely addressed my comments. No further changes requested. The paper needs to be reviewed by the journal's editing staff to smooth additional sentences for English style.
Author Response
Many Thanks
Our Response
The authors have nicely addressed my comments. No further changes requested. The paper needs to be reviewed by the journal's editing staff to smooth additional sentences for English style.
Efforts made to improve the English language.